# Detection of Micro-Doppler Signals of Drones Using Radar Systems with Different Radar Dwell Times

**Jiangkun Gong** [1] , **Jun Yan** [1], **Deren Li** [1] **and Deyong Kong** [2,*]

1   State Key Laboratory of Information Engineering in Surveying, Mapping and Remote Sensing, Wuhan University, Wuhan 430072, China
2   School of Information Engineering, Hubei University of Economics, Wuhan 430205, China
*   Correspondence: kdykong@hbue.edu.cn; Tel.: +86-027-68778527

**Abstract:** Not any radar dwell time of a drone radar is suitable for detecting micro-Doppler (or jet engine modulation, JEM) produced by the rotating blades in radar signals of drones. Theoretically, any X-band drone radar system should detect micro-Doppler of blades because of the micro-Doppler effect and partial resonance effect. Yet, we analyzed radar data detected by three radar systems with different radar dwell times but similar frequency and velocity resolution, including Radar$-\alpha$, Radar$-\beta$, and Radar$-\gamma$ with radar dwell times of 2.7 ms, 20 ms, and 89 ms, respectively. The results indicate that Radar$-\beta$ is the best radar for detecting micro-Doppler (i.e., JEM signals) produced by the rotating blades of a quadrotor drone, DJI Phantom 4, because the detection probability of JEM signals is almost 100%, with approximately 2 peaks, whose magnitudes are similar to that of the body Doppler. In contrast, Radar$-\alpha$ can barely detect any micro-Doppler, and Radar$-\gamma$ detects weak micro-Doppler signals, whose magnitude is only 10% of the body Doppler's. Proper radar dwell time is the key to micro-Doppler detection. This research provides an idea for designing a cognitive micro-Doppler radar by changing radar dwell time for detecting and tracking micro-Doppler signals of drones.

**Keywords:** cognitive micro-Doppler radar; drone detection; Doppler resolution; JEM signals; radar dwell time

## 1. Introduction

Recently, researching topics about using micro-Doppler to detect, classify, and track radar echoes of drones have been hot spots. The most common drones in these studies are drones with rotating blades, such as single-rotor drones, quadrotor drones, six-rotor drones, and even hybrid vertical take-off and landing (VTOL) drones. They are small in size, fly at a slow speed, and are mainly active at low-altitude airspace [1,2]. The rotating movement of rotating blades can modulate the incident radar wave and produce an additional micro-Doppler on the base of the body Doppler contributed by the flying motion of the drone body. Micro-Doppler signals are thought to be useful signatures for radar applications.

Not any one radar system is suitable for detecting and classifying micro-Doppler in radar signals of drones. Currently, both academia and industry have revealed many drone detection radar solutions. They are a wide and diverse variety of types, such as pulse-Doppler marine radar [3], FMCW (frequency-modulated continuous wave) radar [4–7], millimeter-wave radar [8], CW (continuous wave) radar [9], staring radar [10,11], airborne weather radar [12], multistatic radar [13], wide/ultrawideband radar systems [14,15], and even radar networks [16]. Radar vendors also launch commercial off-the-shelf (COTS) counter-drone radars from generation to generation; for example, there are some commercial drone detection radar systems listed in Table 1. Their radar dwell times can be estimated roughly using the rotating rate of the antennas. Generally, with a faster rotating rate comes a shorter radar dwell time. No matter what drone radar is configured, radar dwell time is

one of many factors to extracting both body Doppler signals and micro-Doppler signals of drones in background clutter.

**Table 1.** Some drone detection radars [1].

| Model (Vendor; Country) | Radar Band | Update Rate (Hz) [2] | Range (km) [3] | Identification Strategy [4] |
|---|---|---|---|---|
| Retinar FAR-AD (Meteksan; Turkey) | Ku | 4/15 | 4.4 | Micro-Doppler |
| Gamekeeper 16U (AVEILLANT; UK) | L | 4 | 5 | Micro-Doppler, tracking data. |
| A800(Blighter; UK) | Ku | 1/4 | 3 | Micro-Doppler |
| XENTA-M1 (Weibel; Danish) | X | 1 | 10 | Range-Doppler, micro-Doppler. |
| ReGUARD (Retia; Czech Republic) | X | 1/4 | 6 | Rada cross section (RCS) |
| ELM/2026BF (IAI; Israel) | X | | 5.2 | Tracking data |
| Spyglass™ (Numerica; USA) | Ku | | | Tracking data |
| Gryphon R1400/R1410 (SRC; USA) | X | | 8.5 | Tracking data |
| ELVIRA (Robin; Netherlands) | X | 2/3 | 2.7 | AI, micro-Doppler |
| Giraffe 1X (SAAB; Sweden) | X | 1 | 13 | AI, kinematic, RCS micro-Doppler, etc. |
| GO20 MM (Thales; France) | X | 1/6 | 4 | AI, micro-Doppler |

[1] These data can be found on their official websites. [2] The update rate is the typical value. Some of them can be selectable. [3] The detection range is for drones with RCS of ~0.01 m$^2$, such as DJI Phantom-4. The classification range is normally shorter than the detection range. [4] The specific identification signatures are not available, and those terms are reported in their official brochures.

Generally, the longer the radar dwell time means the better Doppler resolution. Yet, there is still an upper limitation of radar dwell time for a practical radar sensor. First of all, a radar needs a rotating motion with a rotating rate to achieve the 360° cycle-scanning ability. Then, there is a conflict in radar parameter design, in that a rapid rotating update rate and fast beam scanning will result in a short dwell time, and then poor Doppler resolution. The result of cycle-scanning is that the radar dwell time in one scanning cannot be infinitely long. Second, even if a radar can stare in some direction and obtain a long radar dwell time, the micro-Doppler could migrate between different space resolution cells during the long radar dwell time, and then there is an overlap of micro-Doppler either in the range-Doppler cell or time-Doppler cell. Moreover, Radar dwell time not only affects micro-Doppler but also body Doppler. Since micro-Doppler is the additional Doppler around the body Doppler, the ratio of micro-Doppler to body Doppler is also the factor related to the signal extraction. Although some typical high-resolution algorithms have also been investigated for improving Doppler resolution, such as compressed sensing (CS) [17], minimum variance distortionless response (MVDR) [18], multiple signal classification (MUSIC) [19], iterative adaptive algorithm (IAA) [20], and machine learning (ML) technology [21], radar dwell time is still the base factor related to the micro-Doppler. A balance of a suitable radar dwell time is required for extracting and observing micro-Doppler in radar signals of drones.

In this paper, we investigate the proper radar dwell time for detecting the micro-Doppler signals (i.e., jet engine modulation, JEM) modulated by the rotating blades of drones. In Section 2, we discuss the relationship between radar dwell time and the micro-Doppler produced by the rotating blades theoretically and then introduce the three typical radar dwell times of our three radar systems, i.e., Radar$-\alpha$, Radar$-\beta$, and Radar$-\gamma$. In Section 3 and Section 4, we analyze the detection performance of micro-Doppler detected by the three radars and then propose our explanation of the detection results. Finally, we

conclude our point in Section 5. The objectives of this paper include three points. (1) We argue that not any radar dwell time is suitable for detecting micro-Doppler produced by the rotating blades in radar signals of drones, and the proper radar dwell time depends on the rotating period of the blades of drones. (2) We propose that two parameters can be used for evaluating the detection performance of micro-Doppler modulated by drones' blades, including the JEM number and the ratio of the first blade's magnitude to that of the body. (3) We suggest that a cognitive radar can be designed by adjusting the radar dwell time to detect micro-Doppler signals of drones.

## 2. Materials and Methods

### 2.1. Micro-Doppler of Rotating Blades of Drones

Micro-Doppler is the additional Doppler related to the micromotion in addition to the body Doppler. Figure 1a demonstrates the geometry of a radar and a rotating rotor blade of a quadrotor drone. Assume that the azimuth angle of $\alpha$ and the elevation angle of $\beta$ is zero. The shape of a blade is not, and nor can it be seen as, a thin rectangular bar with rotating movement. According to micro-Doppler theory in [22], the received signals of the $k$th blade and its micro-Doppler frequency are given by

$$\left|S_{k,mD}(t)\right| = \sum_{k=0}^{N-1} L sinc\left(\frac{2\pi L}{\lambda}sin\left(\Omega t + \theta_{0,k} + \frac{k2\pi}{N}\right)\right) \tag{1}$$

$$f_{k,mD}(t) = \sum_{k=0}^{N-1}\frac{2\pi L}{\lambda}\Omega\left[-sin\left(\theta_{0,k} + \frac{k2\pi}{N}\right)sin(\Omega t) + cos\left(\theta_{0,k} + \frac{k2\pi}{N}\right)cos(\Omega t)\right] \tag{2}$$

where $N$ is the number of blades, $L$ is the blade length, $\lambda$ is the wavelength, $\Omega$ is the rotation rate, and $\theta_{0,k}$ is the initial rotation angle. Equations (1) and (2) indicate that such micro-Doppler signals are modulated by the rotation rate $\Omega$ through two sinusoidal functions. The maximum values of both $\left|S_{k,mD}(t)\right|$ and $f_{k,mD}(t)$ appear when the direction of the incident wave is perpendicular to the long face of the blade, which means that the rotation angle is 90°, and then we can obtain the "blade flash" signals in the time series and the JEM-like peaks in the spectrum [22–25].

The ideal micro-Doppler or JEM could be simulated when the radar dwell time and frequency resolution are enough. Figure 1b shows the simulated micro-Doppler of rotating blades. The number of blades is 1, and the rotating rate is 100 Hz. The length of a single blade is 0.2 m. The elevation angle is 15°, and the detection range is 20 km. The test band is the X-band, working on 10 GHz. The simulated radar time is 100 ms. Figure 1b shows the flash signals produced by the blade, which are modulated by the rotating rate on the radar images processed by short-time Fourier transform (STFT) algorithm. Figure 1c demonstrates the "blade flash" signals in the time domain with a modulation period of 5 ms and the JEM signals with a frequency interval of about 100 Hz.

Radar dwell time is the key to observing such micro-Doppler. According to Nyquist Theorem, the radar dwell time must be at least longer than twice the rotating period of the rotating blades Given that the rotating rate of the blades of the drone is $\Omega$, the minimum radar dwell time for obtaining such sufficient micro-Doppler is given by

$$T_s = \frac{2}{\Omega} \tag{3}$$

where $T_s$ is the minimum radar dwell time. If the general rotation rate of blades of drones is 100 Hz, the minimum dwell time is approximately 20 ms. Different radar systems have different radar dwell times. If radar dwell time of radar is much shorter than the required one, $T_s$, then we may observe the insufficient micro-Doppler with weaker magnitude and a smaller number of JEM peaks. Furthermore, what happens when the radar dwell time of radar is much longer than the required one? Can we obtain a much clearer micro-Doppler

in the radar signals of drones? What is the best radar dwell time for observing such micro-Doppler?

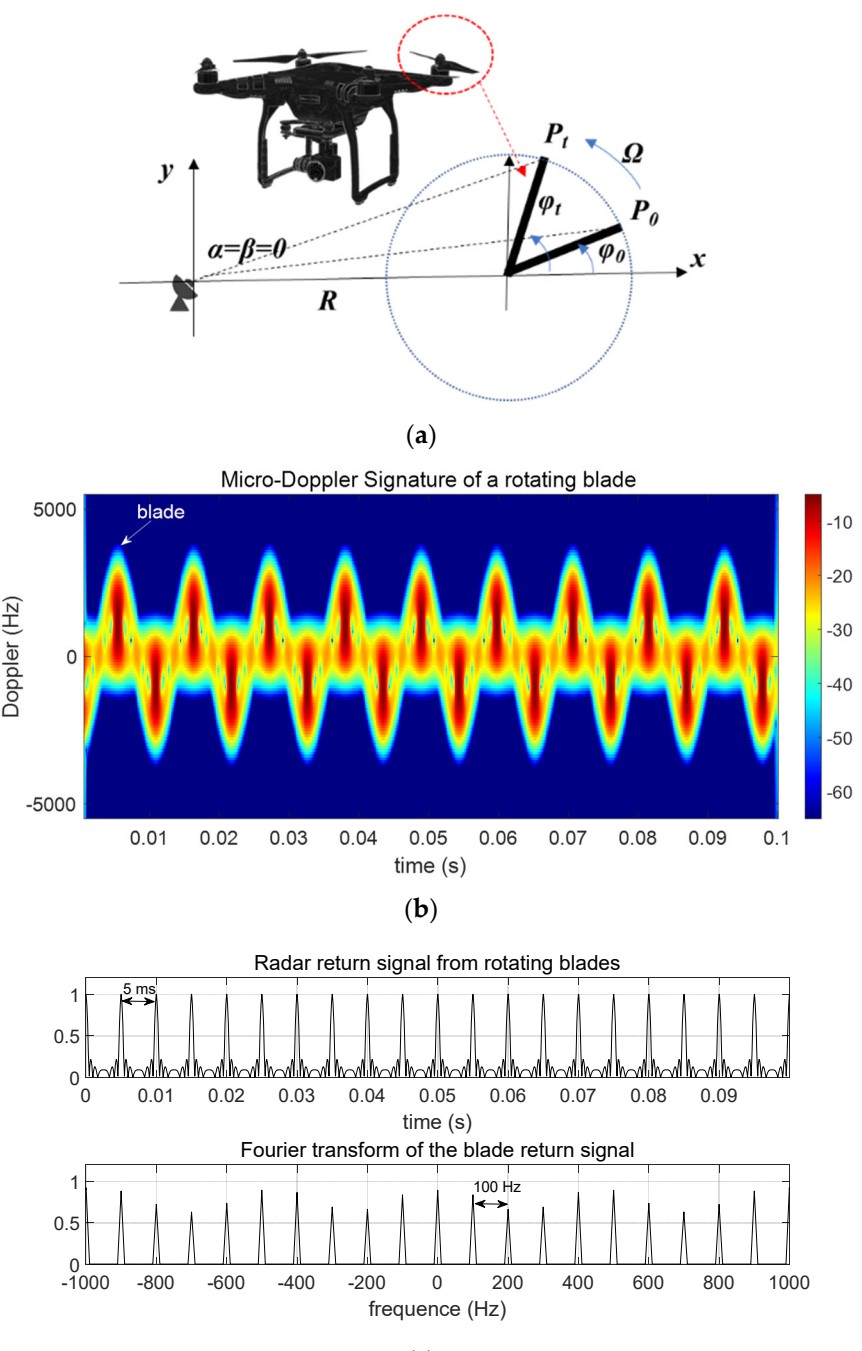

(**a**)

(**b**)

(**c**)

**Figure 1.** Simulated micro-Doppler of rotating blades within X-band data. (**a**) The geometry of the radar and the rotating rotor blades, (**b**) micro-Doppler on the STFT image, (**c**) blade flash and JEM signals.

To evaluate the detection performance of micro-Doppler (i.e., JEM) of rotating blades of drones, we select some parameters about JEM signatures. They are (1) the number of JEM peaks, and (2) the ratio of the first blade's magnitude to the body's magnitude. They can be given, respectively, by

$$N_{mD} = N - 1 \qquad (4)$$

where $N_{mD}$ is the number of JEM peaks, $N$ is the number of Doppler peaks in the spectrum, and the 1 represents the body Doppler (i.e., bulk Doppler); and

$$r_{mb} = \frac{A_{mD}}{A_{bD}} \tag{5}$$

where $r_{mb}$ is the ratio of blade's magnitude to body's magnitude, $A_{mD}$ is the magnitude of the first neighboring blade Doppler, and $A_{bD}$ is the magnitude of the body Doppler. Theoretically, the better detection performance of such micro-Doppler (i.e., JEM) means a bigger $N_{mD}$ and a higher $r_{mb}$.

### 2.2. Experimental Conditions

To investigate the detection performance of such micro-Doppler, a software-defined radar platform that can be changed with radar dwell times as well as other parameters should be used for collecting radar data of drones and then seeking the best radar dwell time for detecting micro-Doppler after investigating the relationship between the radar dwell time and micro-Doppler. Unfortunately, we do not have such resources to conduct this research, and we can only use some radar data of drones detected by three radar systems with typical radar dwell times to explore the first step of such a topic. Instead, we obtained some drone detection radar data detected by three radar systems with typical radar dwell times (i.e., insufficient, moderate, and sufficient). Table 2 lists some parameters of the three radar systems. They are pulse-Doppler radar systems equipped with phased-array antennas. The pulse repetition frequency (PRF) and Doppler resolutions of three radar sensors are similar, but Radar$-\alpha$ (the insufficient one) has the shortest radar dwell time in one coherent pulse interval (CPI) of 2.7 ms, Radar$-\beta$ (the moderate one) has a moderate time of 20 ms, and Radar$-\gamma$ (the sufficient one) has the longest time of 89 ms. Thereby, given the general rotating period (i.e., 10 ms) of the blade, Radar$-\alpha$ can detect insufficient micro-Doppler because the radar dwell time is only 27% of the rotating period, Radar$-\gamma$ can detect sufficient micro-Doppler because the radar dwell time is about 4 times than the rotating period, and Radar$-\beta$ may detect either sufficient or insufficient micro-Doppler based on the rotating rate of the blades in actual cases. The quadcopter drone is a DJI Phantom 4, fabricated by DJI Inc., China. It is a small drone with a flight weight of 1.38 kg. Both its body and propellers are mainly composed of plastic. Its wheelbase is approximately 0.35 m. There are four rotor blades with a length of 0.2 m. The cruise speed is approximately 15 m/s. The maximum flight time is approximately 28 min, with a maximum flight height lower than 500 m. The rotating rate of blades is from 5000 RPM to 7000 RPM (revolutions per minute).

**Table 2.** Parameters of the three drone detection radars.

| Parameters | Radar$-\alpha$ | Radar$-\beta$ | Radar$-\gamma$ |
| --- | --- | --- | --- |
| Radar band | X | X | X |
| CPI (ms) | 2.7 | 20 | 89 |
| PRF (kHz) | 33.3 | 5 | 2.8 |
| Sampling points after zero padding | 2048 | 256 | 256 |
| Frequency resolution (Hz) | 16 | 19 | 11 |
| Doppler resolution (m/s) | 0.163 | 0.285 | 0.165 |
| Range resolution (m) | 3.75 | 12 | 10 |
| Beamwidth | 0.97° | 0.72° | 2° |
| Detection range (m) | 3000 | 10,000 | 6000 |
| Width of the wavefront (m) | 50.7 | 125.6 | 209.4 |
| Space resolution (m$^2$) | 190.1 | 1507.2 | 2094 |
| Radar dwell time per square meter (ms/m$^2$) | 0.014 | 0.013 | 0.042 |

The radars collected these data at three areas. The detection background is mainly ground clutter, but the detection ranges were different. The range of the drones from

Radar$-\alpha$ was about 3 km, the one from Radar$-\beta$ was about 10 km, and that from Radar$-\gamma$ was about 6 km. The drones were flying in the radar beams, and the radars worked in a tracking mode. Thereby, we collected tracking data of drones. The drones were flying in a range widow with a size of 1 km, at an altitude below 300 m. The width of the wavefront at the radar range of a target is calculated by the beamwidth and the detection range, which is

$$W = \theta_{re} R \tag{6}$$

where $W$ is the width of the wavefront, $\theta_{re}$ is the beamwidth, and $R$ is the detection range of the target. Thereby, the spatial resolution of the sector where the target is in can be given approximately by

$$Sp_r = WR_{re} \tag{7}$$

where $Sp_r$ is the spatial resolution and $R_{re}$ is the range resolution. Table 2 also lists the range resolutions, detection ranges, and space resolutions in the three cases. If we divide the spatial resolution by the CPI time, we can obtain the radar dwell time per square meter values of the three radars, which are 0.014 ms/m$^2$, 0.013 ms/m$^2$, and 0.042 ms/m$^2$, respectively. These numbers are similar to each other. Figure 2 shows an example of detecting and tracking the DJI Phantom 4, using Radar$-\gamma$ working in a tracking mode. The blue solid line was the radar beam, and the blue dotted lines were the tracking trace of the drone when flying towards the radar location. We selected some radar data collected in this area and used them in this paper. Similarly, we also collected some data using Radar$-\alpha$ and Radar$-\beta$ in some other areas.

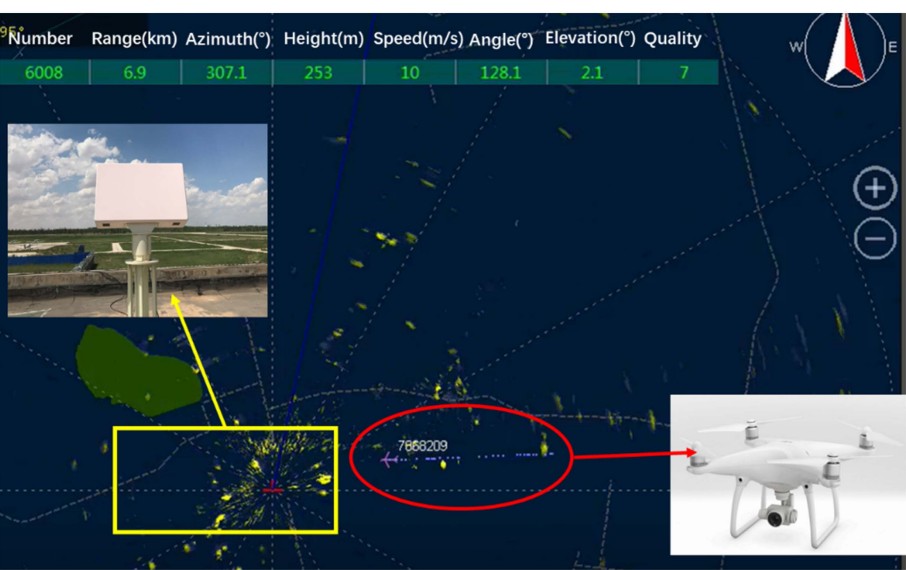

**Figure 2.** Example of tracking trace of the drone on the radar (i.e., Radar$-\gamma$) screenshot.

Although our radar data were collected in a different area, they could still be used for this research for the following reasons. First, we investigated the detection performance of micro-Doppler signals of drones over the radar signals of drones. This means that the radar data of drones were already detected and tracked in the background clutter. Second, we compared the micro-Doppler with the body Doppler; thus, it involves no background clutter. We can also use the normalized magnitudes to remove the interference from the background environment. Thereby, these radar data collected from different ranges in different areas can be used for research to investigate micro-Doppler signals of drones.

## 3. Results

Radar detection is always accompanied by interference from background clutter, and different radar systems with different dwell times can have different performances.

Generally, the longer the radar dwell time, the higher the SNR value of the target. Yet, the noise level of clutter also increases along with the increased radar dwell time. Figure 3 demonstrates radar signals of drones in a similar range window. The black frames in the range-Doppler images mark the range bin that the drone was in. The length values of range windows are 78.75 m, 84 m, and 70 m, respectively. Generally, the ground clutter is mainly around 0 Hz, with different spectral widths, which are 728 Hz, 80 Hz, and 106 Hz when using Radar$-\alpha$, Radar$-\beta$, and Radar$-\gamma$. Although the SNR values of the drone (65.68 dB, 11.43 dB, 11.55 dB) are different in the three cases, its radar echoes can be detected in the clutter. Yet, Radar$-\beta$ can detect the most legible radar signals of the drone because the micro-Doppler signals produced by the rotating blades can also be detected and identified along with the body Doppler (three pots in the black frame in Figure 3b). Other radars seem to detect only body Dopplers of drones on their range-Doppler images (Figure 3a,c). In total, Radar$-\beta$ enjoys a better detection performance for detecting drones than Radar$-\alpha$ and Radar$-\gamma$.

Micro-Doppler signatures of drones produced by the rotating blades are related directly to the radar dwell time. Figure 4 compares the raw radar signals and the spectrums of drones using the three radar systems. The data are extracted from the range windows in Figure 3. The blues words register the Doppler peaks corresponding to either blades or the body of drones. When the drone is flying away from the radar, its Doppler velocity is negative, and when it is flying approaching the radar, the velocity is positive. Each subfigure shows two cases in which the drone flew in a different direction, relative to the radars. According to Table 2, both the frequency resolution and the velocity resolution of the three radars are similar to each other, but the radar dwell times (i.e., 2.7 ms, 20 ms, and 89 ms) are different. The rotating rate of the blades ranges from 5000–7000 RPM (revolutions per minute). According to Formula (3), the minimum radar dwell time to obtain the sufficient micro-Doppler of the rotating blades is 17–24 ms. Therefore, Radar$-\alpha$ can only detect insufficient micro-Doppler signals of the drone, and Radar$-\beta$ observes either sufficient micro-Doppler signals or insufficient micro-Doppler signals based on the rotating rate of the blades, but Radar$-\gamma$ detected only sufficient micro-Doppler signals.

There seems to be only one bulk Doppler (i.e., body Doppler) detected by Radar$-\alpha$, in Figure 4a, which is $-9.4$ m/s (or $+8.6$ m/s). In contrast, several Doppler peaks including one body Doppler and two or three micro-Dopplers appear in the spectrums detected by Radar$-\beta$ and Radar$-\gamma$, which are $-13.2$ m/s, $-8.1$ m/s, $-6$ m/s, $-3.3$ m/s (or 2.1 m/s, 4.2 m/s, 6.9 m/s) in Figure 4b, and $-20.2$ m/s, $-17.0$ m/s, $-13.8$ m/s, $-12.0$ m/s, $-10.8$ m/s (or 6.6 m/s, 9.2 m/s, 11.7 m/s, 14.2 m/s) in Figure 4c. The number of micro-Dopplers (i.e., $N_{mD}$) detected by Radar$-\gamma$ in Figure 4c is about four, and the number detected by Radar$-\beta$ in Figure 4b is about three. Moreover, the ratio of strongest micro-Doppler magnitude to that of body-Doppler (i.e., $r_{mb}$) in Figure 4b is about 1, but this number decreases below 0.2 in Figure 4c. Thereby, Radar$-\gamma$ seems to be able to detect more micro-Dopplers than Radar$-\beta$, but Radar$-\beta$ can detect much stronger micro-Doppler than Radar$-\gamma$. Yet, Radar$-\alpha$ detected the poorest micro-Doppler. This means that a moderate dwell time is better for the micro-Doppler of drones.

It is not proper that the longest radar dwell times come with the best detection performance of micro-Doppler signals. Figure 5 presents the tracking Doppler signals of drones using the three radar systems. The tracking intervals are different from each other. The black dotted curves in Figure 5 describe the changing body Doppler of drones in the three cases. First, similar to the range-Doppler images in Figure 3, there is always clutter when detecting drones, and the Doppler of background clutter mainly stays around 0 m/s. Doppler detection can separate the radar signals of drones from the clutter with small velocities. Second, Radar-$\beta$ can track more enriched micro-Doppler signatures than Radar$-\alpha$ and Radar$-\gamma$. There are always attendant spots around the body Doppler in Figure 5b, which represent the distributed pattern of micro-Doppler modulated by the rotating blades of drones. Yet, these micro-Doppler spots seem to disappear on the images in Figure 5a,c. As we stated in Figure 3, Radar$-\alpha$ cannot detect insufficient micro-Doppler,

and then the micro-Doppler spots disappear in the tracking results. However, Radar$-\gamma$ can detect very strong body Doppler, which is much stronger than micro-Dopplers, and then the micro-Dopplers are suppressed by the body Dopplers and hidden in the images of Figure 5c.

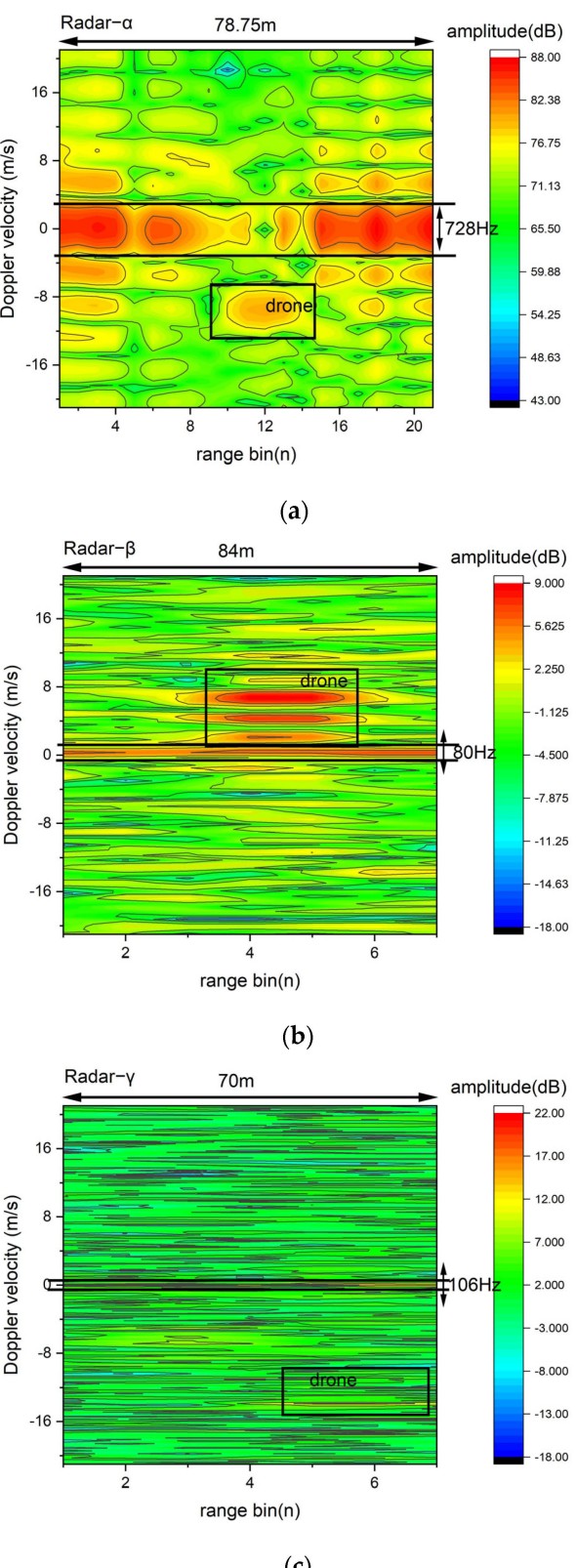

**Figure 3.** Range-Doppler data of drones. (**a**) Radar$-\alpha$, (**b**) Radar$-\beta$, (**c**) Radar$-\gamma$.

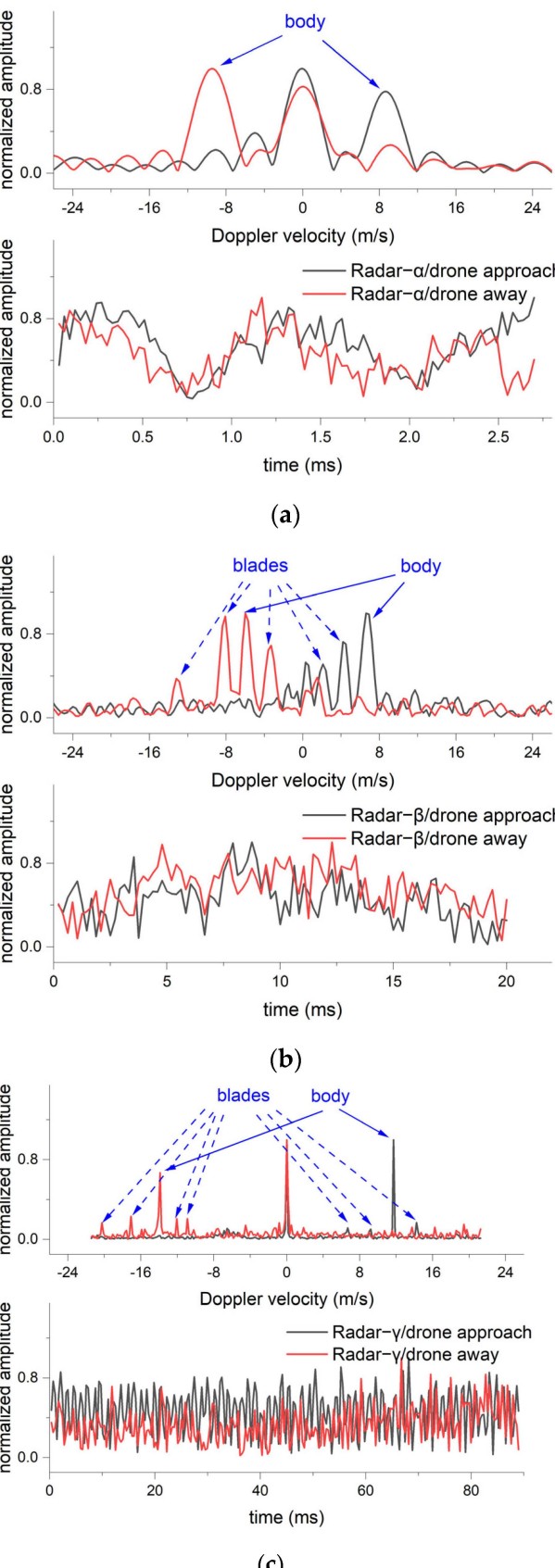

**Figure 4.** Radar signals and spectrums of drones. (**a**) Radar−α, (**b**) Radar−β, (**c**) Radar−γ.

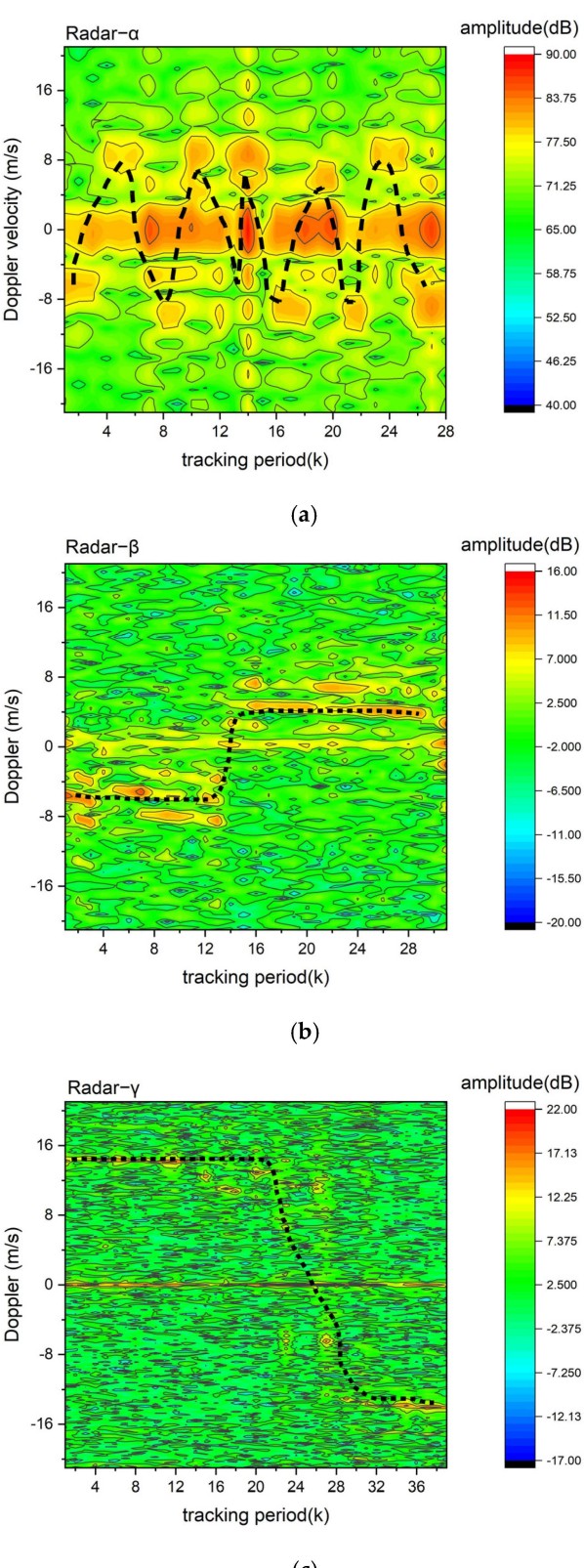

**Figure 5.** Tracking Doppler data of drones. (**a**) Radar−α, (**b**) Radar−β, (**c**) Radar−γ.

The quantification analysis of tracking results indicates that compared to Radar−α and Radar−γ, Radar−β with moderate radar dwell time is a better solution for detecting and tracking micro-Doppler signals of drones among the three radars. Figure 6 demonstrates the tracking parameters of three cases in Figure 5. SNR means signal-to-noise ratio, which

describes the scattering power of a target, and SCR means signal-to-clutter ratio, which presents the scattering superiority of a target to the clutter. The JEM number is the number of micro-Doppler peaks in the spectrum, and blade/body means the ratio of the micro-Doppler's magnitude to the body Doppler's.

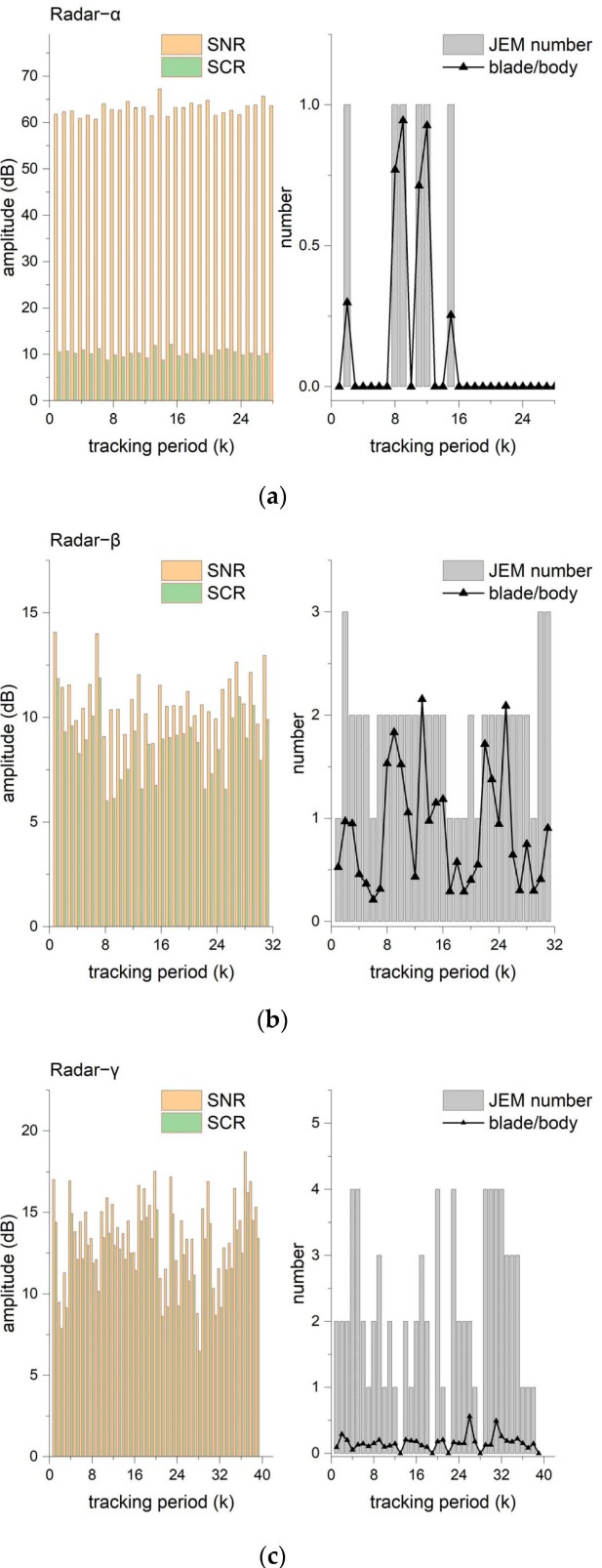

**Figure 6.** Detection results of drones. (**a**) Radar−α, (**b**) Radar−β, (**c**) Radar−γ.

Table [3] shows the statical data of results in Figure [6]. First, both SNR and SCR values are fluctuating in three cases. Since different radar systems have different transmitted power levels and noise levels, the SNR values of drones can fluctuate considerably. For example, the mean SNR of drones detected by Radar$-\alpha$ is 63.01 dB, about six times that of the 10.98 dB detected by Radar$-\beta$. Yet, SCR seems to be much more stable, with only a range of less than 4 dB. To this degree, SCR is a better value for detecting radar signals of targets. Second, Radar$-\alpha$ can detect possible micro-Doppler in some cases (e.g., #2, #8, #9, etc.) and Radar$-\gamma$ sometimes detects no micro-Doppler (e.g., #13, #19, #2, etc.), but Radar$-\beta$ can always detect micro-Doppler in all sampling cases. Thereby, the detection probabilities of JEM detected by the three radars are 21.42%, 100%, and 12.82%, respectively. Third, the numbers of micro-Doppler detected by Radar$-\beta$ and Radar$-\gamma$ are similar at 2, much bigger than the 0.18 detected by Radar$-\alpha$. It means that as long as the micro-Doppler is detected by the radar, there are at least two JEM peaks. Fourth, Radar$-\beta$ can detect the strongest micro-Dopplers with the ratio of blade's signal to body's signal of 0.88, but the number (i.e., 0.16) is very small in the cases detected by Radar$-\gamma$. In some words, even if the micro-Doppler is in the radar signals, it can be neglected by an extraction algorithm. Although the number of cases containing micro-Doppler detected by Radar$-\alpha$ is only 6 among the whole 28, the ratio of blade's signal to body's signal is still about 0.65. Fifth, the frequency offsets between the body's Doppler and the first neighboring blade's Doppler are 469 Hz, 165 Hz, and 158 Hz, respectively.

**Table 3.** Comparison of radar detection of a drone1.

| Contents | Radar$-\alpha$ | Radar$-\beta$ | Radar$-\gamma$ |
|---|---|---|---|
| Detection range (km) | ~3 km | ~10 km | ~6 km |
| Doppler velocity (m/s) | 8.29 | 4.70 | 13.00 |
| Mean SNR (dB) | 63.01 | 10.98 | 14.22 |
| Mean SCR (dB) | 10.23 | 8.71 | 12.17 |
| Probability of JEM signals | 21.42% | 100% | 12.82% |
| Number of JEM peaks | 0.18 | 1.87 | 2.05 |
| The ratio of the blade's magnitude to that of the body | 0.65 | 0.88 | 0.16 |
| Frequency offset between blade and body (Hz) | 469 | 165 | 158 |

## 4. Discussion

Why can Radar$-\beta$ with moderate radar dwell time detect the best micro-Doppler among the three radar systems? Radar dwell time and radar wavelength are the key factors. Micro-Doppler is the additional Doppler related to the micromotion of the microcomponent on the body of a target. For the drone, the blades are the microcomponent, and the rotating motion is the source of the additional Doppler, as shown in Figure [7]. According to Equations (1) and (2), the maximum values of $\left|S_{k,mD}(t)\right|$ and $f_{k,mD}(t)$ appear when the direction of the incident wave is perpendicular to the long face of the blade, which means that the rotation angle is 90°, and then "blade flash" signals appear in the time series, and JEM-like peaks occur in the spectrum. Furthermore, the partial resonance effect will also contribute to the scattering power of blades. It is known that the resonance effect occurs when the sizes of a target (e.g., drones) are comparable with the radar wavelengths, so their scattering properties are calculated via Mie theory [26,27]. As such, the scattering power of the target is an oscillating function of the size, the materials contents, and the wavelength so that the radar reflectivity values can be amplified at another wavelength. Since the wavelength of the X-band is similar to the width of the blade of the drone, and when the transmitted wave is shot directly onto the blade in the direction of perpendicular to the transmission, the partial resonance effect will amplify only the scattering power of the blades. The micro-Doppler effect and the partial resonance effect together cause the strong JEM in the spectrum.

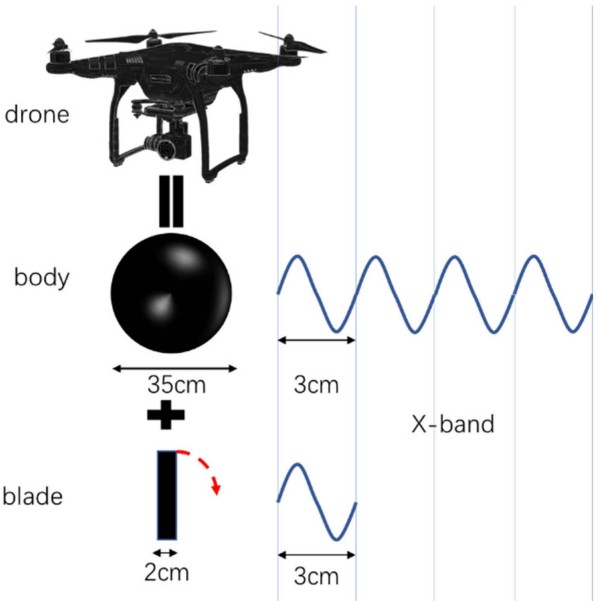

**Figure 7.** Diagram showing radar wave intervals with different structures of a quadrotor drone.

The different coherent integration times (i.e., radar dwell time) on the drone's body and blades also affect the scattering power from the body and the blades. It is known that if the coherent integration is performed and the magnitudes of the returns from all $N$ pulses are added, the SNR of a target increases as $N$. Note that due to the rotating motion of the blades, the real radar dwell times on the blades are always shorter than the body. In our cases, the three radar systems (i.e., Radar$-\alpha$, Radar$-\beta$, Radar$-\gamma$) have radar dwell times of 2.7 ms, 20 ms, and 89 ms, respectively, and the rotating period of the blades of drones is about 10 ms. Only the radar dwell time of 20 ms of Radar$-\beta$ is similar to the two-rotation period of the blades, and then the magnitude of the blade Doppler is similar to that of the body Doppler because of the partial resonance effect. In contrast, the radar dwell time of 2.7 ms of Radar$-\alpha$ is too short, so there is only a 10% probability of "blade flash" signals (or JEM signals) contributed by the micro-Doppler effect and the partial resonance effect. Furthermore, the radar dwell time of 89 ms of Radar$-\gamma$ is about four times the required one. During this period, the blades have eight rotating periods, which means that the magnitude of the blade Doppler is smaller than 1/4 of the body Doppler's magnitude. Thereby, due to improper radar dwell time, Radar$-\alpha$ and Radar$-\gamma$ are not suitable for detecting micro-Doppler of drones.

The promising application of this finding that proper radar dwell time is the key to detecting micro-Doppler of targets is to design the cognitive radar systems detecting micro-Doppler by adjusting the radar dwell time. Cognitive radar systems use adaption between the information extracted from the sensor and the transmission of subsequent illuminating waveforms. A practicable cognitive radar is to use the micro-Doppler information and then enjoy the best detection performance. As shown in Figure 8, a cognitive micro-Doppler drone detection radar can change its radar dwell time and PRF to obtain the best performance in detecting micro-Doppler signals of drones. The radar can learn the transmitted parameters of Radar$-\beta$ and adjust the transmitted parameters by using the JEM signatures including the number of JEM Dopplers, and the ratio of the blade's signal to the body's signal. We believe that this new drone detection radar will work well, like Radar$-\beta$ in this paper. In the future, we will continue to conduct related research, design the cognitive micro-Doppler radar for detecting drones using a software-defined radar platform, and evaluate its performance.

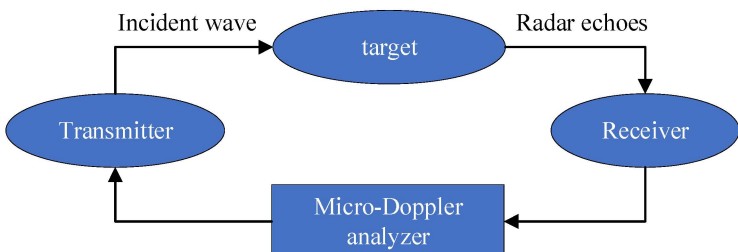

**Figure 8.** Block diagram of a cognitive micro-Doppler radar system.

## 5. Conclusions

The radar dwell time of a drone detection radar is the key to detecting micro-Doppler in radar signals of drones. Theoretically, any X-band drone radar system should detect micro-Doppler signals because of the micro-Doppler effect and partial resonance effect. In this paper, we analyze radar data detected by three radar systems with different radar dwell times and similar frequency and velocity resolution. Radar$-\alpha$, Radar$-\beta$, and Radar$-\gamma$ have radar dwell times of 2.7 ms, 20 ms, and 89 ms, respectively. We use two parameters to evaluate the detection performance of micro-Doppler using the three radar systems, including the number of JEM peaks (See Equation (4)), and the ratio of the first blade's magnitude to the body's magnitude (See Equation (5)). The detection results indicate that Radar$-\beta$ is the best radar for detecting micro-Doppler (i.e., JEM signals) produced by the rotating blades of a quadrotor drone, DJI Phantom 4, because the probability of the JEM signals is almost 100%, with approximately 2 peaks, whose magnitudes are similar to that of the body Doppler. In contrast, Radar$-\alpha$ can barely detect any micro-Doppler, and Radar$-\gamma$ detects weak micro-Doppler signals, whose magnitude is only 10% of the body Doppler's. Furthermore, the best radar dwell time for detecting such micro-Doppler is similar to the two-rotation period of the blades. Our findings demonstrate that micro-Doppler signals could be used for designing a cognitive radar for detecting and tracking micro-Doppler signals of drones.

**Author Contributions:** Conceptualization, J.Y.; methodology, J.G.; software, D.K.; validation, J.Y.; formal analysis, J.G.; investigation, J.Y.; resources, D.L.; data curation, J.Y.; writing—original draft preparation, J.G.; writing—review and editing, D.K..; visualization, D.K.; supervision, J.Y.; project administration, D.L.; funding acquisition, D.K. All authors have read and agreed to the published version of the manuscript.

**Funding:** This research received some support from the Natural Science Foundation of Hubei Providence (General Program: 2021CFB309).

**Institutional Review Board Statement:** Not applicable.

**Informed Consent Statement:** Not applicable.

**Data Availability Statement:** Some of the data presented in this study may be available on request from the corresponding author. The data are not publicly available due to the internal restriction of the research group.

**Acknowledgments:** We appreciate both the testers during the collection of the data, and we also want to thank the authors whose photographs are reproduced in this study. Furthermore, we would like to thank Huiping Hu for her help with processing the figures in this paper.

**Conflicts of Interest:** The authors declare that they have no conflict of interest.

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
