# Peer review of "Detection of Micro-Doppler Signals of Drones Using Radar Systems with Different Radar Dwell Times"

_drones, doi:10.3390/drones6090262_

Round 1
Reviewer 1 Report
Please see the attachment.

Author Response
"Please see the attachment."

Reviewer 2 Report
This manuscript analyzes the performance of three radar systems with different radar dwell times and similar frequency and velocity resolutions in detecting micro-Doppler information in radar signals of drones. The experimental results verify that the dwell time of the radar needs to satisfy the requirements of the Nyquist Theorem. However, I have a few concerns before recommending the acceptance of this paper to MDPI drones. The followings are my several concerns to be clarified.
(1) There is no specific description of the experimental protocol. What mode of operation is the drone in? How does the change in drone flight speed affect the experimental results? What is the configuration plan between the radar and drone? It is suggested that a subsection be added to describe the experimental protocol with a schematic diagram.
(2) The parameters of the three radar systems are listed in Table 1, from which some of their differences can also be observed. Furthermore, the descriptions in lines 138 and 139 appear to indicate that the radars are at different distances from the drone. These differences harm the reliability of the experimental results. Other factors, such as distance, azimuth angle, and elevation angle, should be kept consistent in the experiment to the greatest extent possible. These factors, particularly in a software-defined radar platform, are easier to control than the radar entity. By the way, if this manuscript uses a software platform, it should indicate which platform it is in the appropriate place.
(3) The three radar systems have significantly distinct dwell times. Some intermediate values, such as 30ms, 40ms, 50ms, and 60ms, should be included for further refinement to obtain more illustrative results.
(4) The full text needs to be carefully proofread because there are grammatical problems and clerical errors in the text. The sentence in line 57 is incomplete, as well as Figure 3 in line 169 should be Figure 2.
Author Response
"Please see the attachment."

Round 2
Reviewer 1 Report
The revised manuscript has addressed my concerns. I'd suggest to accept the paper as it is.
Reviewer 2 Report
All my concerns have been resolved. The solution given by this work is insightful and feasible.